# 4 and 7-bit Labeling for Projective and Non-Projective Dependency Trees

**Carlos Gómez-Rodríguez, Diego Roca and David Vilares**
Universidade da Coruña, CITIC
Departamento de Ciencias de la Computación y Tecnologías de la Información
Campus de Elviña s/n, 15071
A Coruña, Spain
{carlos.gomez, d.roca1, david.vilares}@udc.es

## Abstract

We introduce an encoding for syntactic parsing as sequence labeling that can represent any projective dependency tree as a sequence of 4-bit labels, one per word. The bits in each word's label represent (1) whether it is a right or left dependent, (2) whether it is the outermost (left/right) dependent of its parent, (3) whether it has any left children and (4) whether it has any right children. We show that this provides an injective mapping from trees to labels that can be encoded and decoded in linear time. We then define a 7-bit extension that represents an extra plane of arcs, extending the coverage to almost full non-projectivity (over 99.9% empirical arc coverage). Results on a set of diverse treebanks show that our 7-bit encoding obtains substantial accuracy gains over the previously best-performing sequence labeling encodings.

## 1 Introduction

Approaches that cast parsing as sequence labeling have gathered interest as they are simple, fast (Anderson and Gómez-Rodríguez, 2021), highly parallelizable (Amini and Cotterell, 2022) and produce outputs that are easy to feed to other tasks (Wang et al., 2019). Their main ingredient are the encodings that map trees into sequences of one discrete label per word. Thus, various such encodings have been proposed both for constituency (Gómez-Rodríguez and Vilares, 2018; Amini and Cotterell, 2022) and dependency parsing (Strzyz et al., 2019; Lacroix, 2019; Gómez-Rodríguez et al., 2020).

Most such encodings have an unbounded label set, whose cardinality grows with sentence length. An exception for constituent parsing is tetratagging (Kitaev and Klein, 2020). For dependency parsing, to our knowledge, no bounded encodings were known. Simultaneously to this work, Amini et al. (2023) have just proposed one: hexatagging, where projective dependency trees are represented by tagging each word with one of a set of 8 tags.[1]

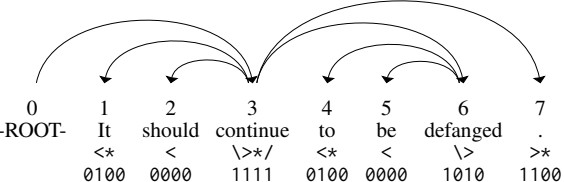

Figure 1: A dependency tree and its 4-bit encoding.

**Contribution** We present a bounded sequence-labeling encoding that represents any projective dependency tree with 4 bits (i.e., 16 distinct labels) per word. While this requires one more bit than hexatagging, it is arguably more straightforward, as the bits directly reflect properties of each node in the dependency tree without an intermediate constituent structure, as hexatagging requires. Also, it has a clear relation to existing bracketing encodings, and has a straightforward non-projective extension using 7 bits with almost full non-projective coverage. Empirical results show that our encoding provides more accurate parsers than the existing unbounded bracketing encodings, which had the best previous results among sequence-labeling encodings, although it underperforms hexatagging.

## 2 Projective Encoding

Let $T_n$ be a set of unlabeled dependency trees[2] for sentences of length $n$. A sequence-labeling encoding defines a function $\Phi_n : T_n \to L^n$, for a label set $L$. Thus, each tree for a sentence $w_1 \ldots w_n$ is encoded as a sequence of labels, $l_1 \ldots l_n$, that assigns a label $l_i \in L$ to each word $w_i$.

We define the *4-bit projective encoding* as an encoding where $T_n$ is the set of projective depen-

---

[1]The "hexa" in the name comes from a set of six atoms

used to define the labels. However, the label for each word is composed of two such atoms (one from a set of two, and other from a set of four) so there are eight possible labels per word.

[2]For simplicity, we ignore dependency labels in definitions. In the implementation, they are added as a separate component to the label of each word, following common practice.

dency trees, and we assign to each word $w_i$ a label $l_i = b_0 b_1 b_2 b_3$, such that $b_j$ is a boolean as follows:

- $b_0$ is true if $w_i$ is a right dependent, and false if it is a left dependent. Root nodes are considered right dependents for this purpose (i.e., we assume that they are linked as dependents of a dummy root node $w_0$ located to the left).

- $b_1$ is true iff $w_i$ is the outermost right (or left) dependent of its parent node.

- $b_2$ (respectively, $b_3$) is true iff $w_i$ has one or more left (right) dependents.

All combinations of the four bits are possible, so we have 16 possible labels.

For easier visualization and comparison to existing bracketing encodings, we will represent the values of $b_0$ as > (right dependent) or < (left dependent), $b_1$ as * (true) or blank (false), and $b_2$ and $b_3$ respectively as \ and / (true) or blank (false). We will use these representations with set notation to make claims about a label's bits, e.g. >* $\in l$ means that label $l$ has $b_0 = 1, b_1 = 1$. Figure 1 shows a sample tree encoded with this method.

We will now show how to encode and decode trees, and prove that the encoding is a total, injective map from projective trees to label sequences.

**Encoding and Totality**   Encoding a tree is trivial: one just needs to traverse each word and apply the definition of each bit to obtain the label. This also means that our encoding from trees to labels is a total function, as the labels are well defined for any dependency tree (and thus, for any projective tree).

**Decoding and Injectivity**   Assuming a well-formed sequence of labels, we can decode it to a tree. We can partition the arcs of any tree $t \in T_n$ into a subset of left arcs, $t_l$, and a subset of right arcs, $t_r$. We will decode these subsets separately. Algorithm 1 shows how to obtain the arcs of $t_r$.

The idea of the algorithm is as follows: we read labels from left to right. When we find a label containing /, we know that the corresponding node will be a source of one or more right arcs. We push it into the stack. When we find a label with >, we know that its node is the target of a right arc, so we link it to the / on top of the stack. Additionally, if the label contains *, the node is a rightmost sibling, so we pop the stack because no more arcs will be

---

**Algorithm 1** To decode right arcs in the 4-bit encoding.
```
 1: function DECODERIGHTARCS(l_1..l_n)
 2:     s ← empty stack
 3:     a ← empty set of arcs
 4:     s.push(0)              ▷ corresponding to dummy root
 5:     for i ← 1 to n do
 6:         if >∈ l_i then
 7:             a.addArc( s.peek() → i)
 8:             if * ∈ l_i then
 9:                 s.pop()
10:             end if
11:         end if
12:         if / ∈ l_i then
13:             s.push(i)
14:         end if
15:     end for
16:     return a
17: end function
```

---

created from the same head. Otherwise, we do not pop as we expect more arcs from the same origin.[3]

Intuitively, this lets us generate all the possible non-crossing combinations of right arcs: the stack enforces projectivity (to cover a / label with a dependency we need to remove it from the stack, so crossing arcs from inside the covering dependency to its right are not allowed), and the distinction between > with and without * allows us to link a new node to any of the previous, non-covered nodes.

To decode left arcs, we use a symmetric algorithm DecodeLeftArcs (not shown as it is analogous), which traverses the labels from right to left, operating on the elements \ and < rather than / and >; with the difference that the stack is not initialized with the dummy root node (as the arc originating in it is a right arc). By the same reasoning as above, this algorithm can obtain all the possible non-crossing configurations of left arcs, and hence the mapping is injective. The decoding is trivially linear-time with respect to sequence length.

A sketch of an injectivity proof can be based on showing that the set of right arcs generated by Algorithm 1 (and the analogous for left arcs) is the only possible one that meets the conditions of the labels and does not have crossing arcs (hence, we cannot have two projective trees with the same encoding). To prove this, we can show that at each iteration, the arc added by line 7 of Algorithm 1 is the only possible alternative that can lead to a legal projective tree (i.e., that s.peek() is the only possible parent of node $i$). This is true because (1)

---

[3]Note that, thus, the interpretation of the symbols / and > is similar to that of the same symbols in the unbounded bracketing encoding by (Strzyz et al., 2019), but here the / symbol is acting as a "superbracket" that matches several opposing brackets (Yli-Jyrä, 2017; Yli-Jyrä, 2019)

if we choose a parent to the left of `s.peek()`, then
we cover `s.peek()` with a dependency, while it has
not yet found all of its right dependents (as other-
wise it would have been popped from the stack),
so a crossing arc will be generated later; (2) if we
choose a parent to the right of `s.peek()` and to the
left of $i$, its label must contain / (otherwise, by def-
inition, it could not have right dependents) and not
be on the stack (as the stack is always ordered from
left to right), so it must have been removed from
the stack due to finding all its right dependents, and
adding one more would violate the conditions of
the encoding; and finally (3) a parent to the right of
$i$ cannot be chosen as the algorithm is only consid-
ering right arcs. Together with the analogous proof
for the symmetric algorithm, we show injectivity.

**Coverage** While we have defined and proved this
encoding for projective trees,[4] its coverage is ac-
tually larger: it can encode any dependency forest
(i.e., does not require connectedness) such that arcs
in the same direction do not cross (i.e., it can han-
dle some non-projective structures where arcs only
cross in opposite directions, as the process of en-
coding and decoding left and right arcs is indepen-
dent). This is just like in the unbounded bracketing
encodings of (Strzyz et al., 2019), but this extra
coverage is not very large in practice, and we will
define a better non-projective extension later.

**Non-surjectivity** Just like other sequence-
labeling encodings (Strzyz et al., 2019; Lacroix,
2019; Strzyz et al., 2020, inter alia), ours is not
surjective: not every label sequence corresponds to
a valid tree, so heuristics are needed to fix cases
where the sequence labeling component generates
an invalid sequence. This can happen regardless
of whether we only consider a tree to be valid if
it is projective, or we accept the extra coverage
mentioned above. For example, a sequence where
the last word is marked as a left child (<) is invalid
in either case. Trying to decode invalid label
sequences will result in trying to pop an empty
stack or leaving material in the stack after finishing
Algorithm 1 or its symmetric. In practice, we can

---

[4]If we did not consider a dummy root, we would be able
to cover planar trees, rather than just projective trees (as in the
bracketing of (Strzyz et al., 2019)), but this would require an
extra label for the sentence's syntactic root. Instead, we use a
dummy root on the left and explicitly encode the arc from it to
the syntactic root, which is thus labeled as a right child instead
of using an extra label. This simplifies the encoding, and the
practical difference between the coverage of projectivity and
planarity is small (Gómez-Rodríguez and Nivre, 2013).

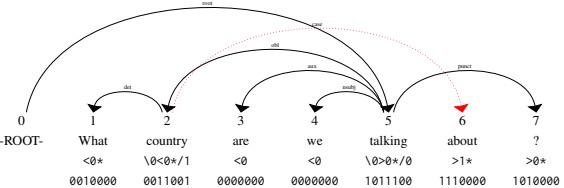

Figure 2: A non-projective tree and its 7-bit encoding.

skip dependency creation when the stack is empty,
ignore material left in the stack after decoding,
break cycles and (if we require connectedness)
attach any unconnected nodes to a neighbor.

## 3 Non-Projective Encoding

For a wider coverage of non-projective dependency
trees (including the overwhelming majority of trees
found in treebanks), we use the same technique as
defined for unbounded brackets in (Strzyz et al.,
2020): we partition dependency trees into two sub-
sets (planes) of arcs (details in Appendix D), and
this lets us define a *7-bit non-projective encoding*
by assigning each word $w_i$ a label $l_i = (b_0 \ldots b_6)$,
where:

- $b_0 b_1$ can take values <0 ($w_i$ is a left dependent
  in the first plane), >0 (right dependent in the
  1st plane), <1 or >1 (same for the 2nd plane).

- $b_2$ is true iff $w_i$ is the outermost right (or left)
  dependent of its parent (regardless of plane).
  We represent it as * if true or blank if false.

- $b_3$ (respectively, $b_4$) is true iff $w_i$ has one or
  more left (right) dependents in the first plane.
  We denote it as \0 (/0) if true, blank if false.

- $b_5$ and $b_6$ are analogous to $b_3$ and $b_4$, but in
  the second plane, represented as \1 or /1.

Every 7-bit combination is possible, leading to
128 distinct labels. Figure 2 shows an example of a
non-projective tree represented with this encoding.

The encoding is able to cover every possible
dependency tree whose arc set can be partitioned
into two subsets (planes), such that arcs with the
same direction and plane do not cross.

This immediately follows from defining the de-
coding with a set of four algorithms, two for de-
coding left and right arcs on the first plane (defined
as Algorithm 1 and its symmetric, but consider-
ing only the symbols making reference to arcs in
the first plane) and other two identical decoding
passes for the second plane. With this, injectivity

| Treebank | B | | B-2P | | 4bit | | 7bit | |
|---|---|---|---|---|---|---|---|---|
| | L | C | L | C | L | C | L | C |
| PTB | 114 | >99.99 | 124 | 100 | 16 | >99.99 | 28 | 100 |
| Russian_GSD | 104 | 99.76 | 166 | >99.99 | 16 | 99.61 | 70 | >99.99 |
| Finnish_TDT | 121 | 99.72 | 172 | >99.99 | 16 | 99.35 | 65 | >99.99 |
| Anc-Greek_Perseus | 259 | 95.81 | 527 | 99.24 | 16 | 88.93 | 128 | 99.24 |
| Chinese_GSD | 101 | 99.91 | 152 | >99.99 | 16 | 99.84 | 46 | >99.99 |
| Hebrew_HTB | 97 | 99.98 | 125 | 100 | 16 | 99.98 | 36 | 100 |
| Tamil_TTB | 51 | 99.94 | 58 | 100 | 16 | 99.98 | 22 | 100 |
| Uyghur_UDT | 78 | 99.43 | 150 | >99.99 | 16 | 99.85 | 58 | >99.99 |
| Wolof_WTB | 74 | 99.83 | 111 | >99.99 | 16 | 99.06 | 46 | >99.99 |
| English_EWT | 110 | 99.88 | 174 | >99.99 | 16 | 99.75 | 63 | >99.99 |
| Macro average | 110.9 | 99.43 | 165.9 | 99.92 | 16 | 98.62 | 56.2 | 99.92 |

Table 1: Number of labels (L) and coverage (C) for each treebank and encoding. B and B-2P are the baselines.

is shown in the same way as for the 4-bit encoding. Decoding is still linear-time.

Note that the set of trees covered by the encoding, described above, is a variant of the set of 2-Planar trees (Yli-Jyrä, 2003; Gómez-Rodríguez and Nivre, 2010), which are trees that can be split into two planes such that arcs within the same plane do not cross, regardless of direction. Compared to 2-Planar trees, and just like the encodings in (Strzyz et al., 2020), our set is extended as it allows arcs with opposite directions to cross within the same plane. However, it also loses some trees because the dummy root arc is also counted when restricting crossings, whereas in 2-Planar trees it is ignored.

## 4 Experiments

We compare our 4-bit and 7-bit encodings to their unbounded analogs, the bracketing (Strzyz et al., 2019) and 2-planar bracketing encodings (Strzyz et al., 2020) which overall are the best performing in previous work (Muñoz-Ortiz et al., 2021). We use MaChAmp (van der Goot et al., 2021) as a sequence labeling library, with default hyperparameters (Appendix B). We use XLM-RoBERTa (Conneau et al., 2020) followed by two separate one-layered feed-forward networks, one for syntactic labels and another for dependency types. We evaluate on the Penn Treebank Stanford Dependencies 3.3.0 conversion and on UD 2.9: a set of 9 linguistically-diverse treebanks taken from (Anderson and Gómez-Rodríguez, 2020), and a low-resource set of 7 (Anderson et al., 2021). We consider multiple subsets of treebanks as a single subset could be fragile (Alonso-Alonso et al., 2022).

Table 1 compares the compactness of the encodings by showing the number of unique syntactic labels needed to encode the (unlabeled) trees in the training set (i.e. the label set of the first task). The new encodings yield clearly smaller label set sizes,

| Treebank | B | B-2P | 4bit | 7bit |
|---|---|---|---|---|
| PTB | 94.62 | 92.03 | **94.72** | 94.66 |
| Russian_GSD | 87.84 | 87.36 | 88.04 | **89.58** |
| Finnish_TDT | 92.45 | 92.37 | 92.19 | **92.74** |
| Anc-Greek_Perseus | 71.84 | 71.76 | 67.63 | **75.36** |
| Chinese_GSD | 85.23 | 84.38 | 85.36 | **85.70** |
| Hebrew_HTB | 90.25 | 90.21 | **90.81** | 90.58 |
| Tamil_TTB | 63.65 | 61.68 | 65.16 | **65.69** |
| Uyghur_UDT | 67.22 | 65.49 | 67.17 | **69.10** |
| Wolof_WTF | 75.04 | 74.59 | **76.24** | 75.57 |
| English_EWT | 91.03 | 91.30 | 89.48 | **91.78** |
| Macro average | 81.92 | 81.12 | 81.68 | **83.08** |

Table 2: LAS for the linguistically-diverse test sets

| Treebank | B | B-2P | 4bit | 7bit |
|---|---|---|---|---|
| Belarusian_HSE | 85.21 | 86.83 | 86.77 | **88.23** |
| Galician_TreeGal | 78.32 | 77.94 | **81.54** | 81.22 |
| Lithuanian_HSE | 52.26 | 49.53 | 55.56 | **56.02** |
| Marathi_UFAL | 62.13 | 55.19 | 66.50 | **67.19** |
| Old-East-Slavic_RNC | 64.15 | 63.43 | **68.96** | 68.84 |
| Welsh_CCG | 81.17 | 80.91 | **82.31** | 82.00 |
| Tamil_TTB | 63.65 | 61.68 | 65.16 | **65.69** |
| Macro average | 69.56 | 67.93 | 72.40 | **72.74** |

Table 3: LAS for the low-resource test sets

as predicted in theory. In particular, the 4-bit encoding always uses its 16 distinct labels. The 7-bit encoding only needs its theoretical maximum of 128 labels for the Ancient Greek treebank (the most non-projective one). On average, it uses around a third as many labels as the 2-planar bracketing encoding, and half as many as the basic bracketing. Regarding coverage, the 7-bit encoding covers over 99.9% of arcs, like the 2-planar bracketing. The 4-bit encoding has lower coverage than basic brackets: both cover all projective trees, but they differ on coverage of non-projectivity (see Appendix C for an explanation of the reasons). More detailed data (e.g. coverage and label set size for low-resource treebanks) is in Appendix A.

Table 2 shows the models' performance in terms of LAS. The 4-bit encoding has mixed performance, excelling in highly projective treebanks like the PTB or Hebrew-HTB, but falling behind in non-projective ones like Ancient Greek, which is consistent with the lower non-projective coverage. The 7-bit encoding, however, does not exhibit this problem (given the almost total arc coverage mentioned above) and it outperforms both baselines for every treebank: the basic bracketing by 1.16 and the 2-planar one by 1.96 LAS points on average.[5]

If we focus on low-resource corpora (Table 3), label set sparsity is especially relevant so compact-

---

[5]In our setup, the basic bracketing encoding baseline beats the 2-planar baseline on average, contrary to the results in the papers that introduced them (cf. (Strzyz et al., 2020)). This likely owes to the architecture used: they used BiLSTMs, whereas we perform our experiments on a more competitive architecture with XLM-RoBERTa.

ness further boosts accuracy. The new encodings obtain large improvements, the 7-bit one surpassing the best baseline by over 3 average LAS points.

## 4.1 Additional results: splitting bits and external parsers

We perform additional experiments to test implementation variants of our encodings, as well as to put our results into context with respect to non-sequence-labeling parsers and simultaneous work. In the previous tables, both for the 4-bit and 7-bit experiments, all bits were predicted as a single, atomic task. We contrast this with a multi-task version where we split certain groups of bits to be predicted separately. We only explore a preliminary division of bits. For the 4-bit encoding, instead of predicting a label of the form $b_0b_1b_2b_3$, the model predicts two labels of the form $b_0b_1$ and $b_2b_3$, respectively. We call this method 4-bit-s. For the 7-bit encoding, we decided to predict the bits corresponding to each plane as a separate task, i.e., $b_0b_2b_3b_4$ and $b_1b_5b_6$. We call this method 7-bit-s. We acknowledge that other divisions could be better. However, this falls outside the scope of this paper.

We additionally compare our results with other relevant models. As mentioned earlier, alongside this work, Amini et al. (2023) introduced a parsing-as-tagging method called hexatagging. In what follows, we abbreviate this method as 6tg. We implement 6tg under the same framework as our encodings for homogeneous comparison, and we predict these hexatags through two separate linear layers, one to predict the arc representation and another for the dependency type. We also consider a split version, 6tg-s, where the two components of the arc representation are predicted separately. For a better understanding of their method, we refer the reader to Amini et al. and Appendix E. Finally, we include a comparison against the biaffine graph-based parser by Dozat et al. (2017). For this, we trained the implementation in SuPar[6] using xlm-roberta-large as the encoder, which is often taken as a strong upper bound baseline.

Table 4 compares the performance of external parsers with our bit encodings. First, the results show that the choice of whether to split labels into components or not has a considerable influence, both for 6tg (where splitting is harmful across the board) and for our encodings (where it is mostly

---
[6] https://github.com/yzhangcs/parser

| Treebank | 4-bit | 7-bit | 6tg | 6tg-s | 4-bit-s | 7-bit-s | biaffine |
|---|---|---|---|---|---|---|---|
| PTB | 94.72 | 94.66 | **96.13** | 96.04 | 94.92 | 94.88 | 95.32 |
| Russian$_{GSD}$ | 88.04 | 89.58 | **91.83** | 90.95 | 88.78 | 90.18 | 90.17 |
| Finnish$_{TDT}$ | 92.19 | 92.74 | **94.12** | 92.66 | 92.11 | 93.10 | 93.33 |
| Anc-Greek$_{Perseus}$ | 67.63 | 75.36 | 73.12 | 72.78 | 68.02 | 76.12 | **79.81** |
| Chinese$_{GSD}$ | 85.36 | 85.70 | 87.39 | 87.32 | 85.99 | 86.13 | **88.67** |
| Hebrew$_{HTB}$ | 90.81 | 90.58 | **92.82** | 91.27 | 90.81 | 91.05 | 91.88 |
| Tamil$_{TTB}$ | 65.16 | 65.69 | **78.33** | 76.32 | 66.99 | 67.19 | 67.52 |
| Uyghur$_{UDT}$ | 67.17 | 69.10 | 71.11 | 65.23 | 67.55 | 69.13 | **72.33** |
| Wolof$_{WTF}$ | 76.24 | 75.57 | 76.04 | 72.11 | **76.85** | 76.24 | 76.73 |
| English$_{EWT}$ | 89.48 | 91.78 | 92.62 | 90.06 | 89.48 | 92.15 | **92.72** |
| Macro avg | 81.68 | 83.08 | **85.35** | 83.47 | 82.15 | 83.62 | 84.85 |

Table 4: LAS comparison against related parsers, for the linguistically-diverse test sets.

| Treebank | 4bit | 7bit | 6tg | 6tg-s | 4bit-s | 7bit-s | biaffine |
|---|---|---|---|---|---|---|---|
| Belarusian$_{HSE}$ | 86.77 | 88.23 | 89.14 | 89.01 | 87.01 | 88.52 | **93.83** |
| Galician$_{TreeGal}$ | 81.54 | 81.22 | 82.03 | 81.94 | 81.97 | 81.31 | **86.81** |
| Lithuanian$_{HSE}$ | 55.56 | 56.02 | 64.47 | **64.74** | 55.97 | 57.31 | 56.75 |
| Marathi$_{UFAL}$ | 66.50 | 67.19 | **75.00** | 74.66 | 66.92 | 67.57 | 61.22 |
| Old-East-Slavic$_{RNC}$ | 68.96 | 68.84 | 71.35 | 71.37 | 69.02 | 68.86 | **72.06** |
| Welsh$_{CCG}$ | 82.31 | 82.00 | **87.05** | 86.92 | 82.62 | 82.13 | 85.05 |
| Tamil$_{TTB}$ | 65.16 | 65.69 | **78.33** | 77.91 | 65.27 | 65.82 | 76.12 |
| Macro average | 72.40 | 72.74 | **78.19** | 78.07 | 72.68 | 73.07 | 75.97 |

Table 5: LAS comparison against related parsers, for the low-resource test sets.

beneficial, perhaps because the structure of the encoding in bits with independent meanings naturally lends itself to multi-task learning). Second, on average, the best (multi-task) version of our 7-bit encoding is about 1.7 points behind the 6tg and 1.2 behind biaffine state-of-the-art parsers in terms of LAS. However, the difference between versions with and without multi-task learning suggests that there might be room for improvement by investigating different splitting techniques. Additionally, in Appendix F, Table 14 compares the processing speeds of these parsers (on a single CPU). In Appendix G, Tables 15 and 16 show how often heuristics are applied in decoding.

Finally, Table 5 shows the external comparison on the low-resource treebanks, where our encodings lag further behind biaffine and especially 6tg, which surpasses 7-bit-s by over 5 points.

## 5 Conclusion

We have presented two new bracketing encodings for dependency parsing as sequence labeling, which use a bounded number of labels. The 4-bit encoding, designed for projective trees, excels in projective treebanks and low-resource setups. The 7-bit encoding, designed to accommodate non-projectivity, clearly outperforms the best prior sequence-labeling encodings across a diverse set of treebanks. The source code is available at https://github.com/Polifack/CoDeLin/releases/tag/1.25.

## Limitations

In our experiments, we do not perform any hyperparameter optimization or other task-specific tweaks to try to bring the raw accuracy figures as close as possible to state of the art. This is for several reasons: (1) limited resources, (2) the paper having a mainly theoretical focus, with the experiments serving to demonstrate that our encodings are useful when compared to alternatives (the baselines) rather than chasing state-of-the-art accuracy, and (3) because we believe that one of the primary advantages of parsing as sequence labeling is its ease of use for practitioners, as one can perform parsing with any off-the-shelf sequence labeling library, and our results directly reflect this kind of usage. We note that, even under such a setup, raw accuracies are remarkably good.

## Ethics Statement

This is a primarily theoretical paper that presents new encodings for the well-known task of dependency parsing. We conduct experiments with the sole purpose of evaluating the new encodings, and we use publicly-available standard datasets that have long been in wide use among the NLP community. Hence, we do not think this paper raises any ethical concern.

## Acknowledgments

This work has received funding by the European Research Council (ERC), under the Horizon Europe research and innovation programme (SALSA, grant agreement No 101100615), ERDF/MICINN-AEI (SCANNER-UDC, PID2020-113230RB-C21), Xunta de Galicia (ED431C 2020/11), Grant GAP (PID2022-139308OA-I00) funded by MCIN/AEI/10.13039/501100011033/ and by ERDF "A way of making Europe", and Centro de Investigación de Galicia "CITIC", funded by the Xunta de Galicia through the collaboration agreement between the Consellería de Cultura, Educación, Formación Profesional e Universidades and the Galician universities for the reinforcement of the research centres of the Galician University System (CIGUS).

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

## A   Further Data

Tables 6 and 7 show treebank statistics for the general and low-resource set of treebanks, respectively.

| Treebank | projective | 1-planar | r arcs | avg d |
|---|---|---|---|---|
| PTB | 99.89% | 99.89% | 48.74% | 2.295 |
| Russian$_{GSD}$ | 93.87% | 93.89% | 49.03% | 2.263 |
| Finnish$_{TDT}$ | 93.85% | 93.88% | 52.88% | 2.365 |
| Anc-Greek$_{Perseus}$ | 37.66% | 37.67% | 52.81% | 2.447 |
| Chinese$_{GSD}$ | 97.75% | 97.87% | 63.67% | 2.440 |
| Hebrew$_{HTB}$ | 96.26% | 96.28% | 49.21% | 2.242 |
| Tamil$_{TTB}$ | 98.33% | 98.33% | 68.56% | 2.262 |
| Uyghur$_{UDT}$ | 95.02% | 96.03% | 64.31% | 2.140 |
| Wolof$_{WTF}$ | 97.01% | 97.10% | 48.21% | 2.519 |
| English$_{EWT}$ | 97.47% | 97.63% | 57.18% | 2.525 |

Table 6: Statistics for the linguistically-diverse set of treebanks: percentage of projective trees, 1-planar trees, percentage of rightward arcs (r arcs), and average dependency distance (avg d).

| Treebank | projective | 1-planar | r arcs | avg d |
|---|---|---|---|---|
| Belarusian$_{HSE}$ | 94.92% | 95.22% | 46.92% | 2.232 |
| Galician$_{TreeGal}$ | 88.80% | 89.20% | 53.02% | 2.530 |
| Lithuanian$_{HSE}$ | 85.93% | 86.69% | 58.40% | 2.321 |
| Old-East-Slavic$_{RNC}$ | 66.26% | 66.35% | 58.21% | 2.433 |
| Marathi$_{UFAL}$ | 95.92% | 96.35% | 50.81% | 2.362 |
| Welsh$_{CCG}$ | 98.24% | 98.24% | 43.94% | 2.324 |
| Tamil$_{TTB}$ | 98.33% | 98.33% | 68.56% | 2.262 |

Table 7: Statistics for the low-resource set of treebanks: percentage of projective trees, 1-planar trees, percentage of rightward arcs (r arcs), and average dependency distance (avg d).

Table 8 shows the number of labels and the arc coverage of each considered encoding for the low-resource treebank set of Anderson et al. (2021), in the same notation as in Table 1. As can be seen in the table, the trends are analogous to those for the other treebanks (Table 1 in the main text).

| Treebank | B | | B-2P | | 4bit | | 7bit | |
|---|---|---|---|---|---|---|---|---|
| | L | C | L | C | L | C | L | C |
| Belarusian$_{HSE}$ | 133 | 99.53 | 228 | >99.99 | 16 | 99.46 | 89 | >99.99 |
| Galician$_{TreeGal}$ | 79 | 99.51 | 129 | >99.99 | 16 | 99.52 | 60 | >99.99 |
| Lithuanian$_{HSE}$ | 64 | 98.88 | 84 | 99.98 | 16 | 98.82 | 45 | 99.98 |
| Marathi$_{UFAL}$ | 46 | 99.44 | 58 | 100 | 16 | 99.32 | 36 | 100 |
| Old-East-Slavic$_{RNC}$ | 134 | 97.66 | 230 | 99.94 | 16 | 97.46 | 86 | 99.94 |
| Welsh$_{CCG}$ | 53 | 99.90 | 71 | 100 | 16 | 99.93 | 38 | 100 |
| Tamil$_{TTB}$ | 51 | 99.82 | 58 | 100 | 16 | 99.84 | 22 | 100 |
| Macro average | 80.0 | 99.25 | 122.6 | 99.99 | 16 | 99.19 | 53.7 | 99.99 |

Table 8: Number of labels (L) and arc coverage (C) for each low-resource treebank and encoding. B and B-2P are the baselines.

Tables 9 and 10 show the coverage of the encodings in terms of full trees, rather than arcs (i.e., what percentage of the dependency trees in each treebank can be fully encoded and decoded back by each of the encodings).

| Treebank | B | B-2P | 4bit | 7bit |
|---|---|---|---|---|
| PTB | >99.99% | 100% | >99.99% | 100% |
| Russian$_{GSD}$ | 96.94% | 99.92% | 95.65% | 99.92% |
| Finnish$_{TDT}$ | 99.43% | 100% | 99.35% | 100% |
| Anc-Greek$_{Perseus}$ | 72.25% | 90.63% | 50.48% | 90.63% |
| Chinese$_{GSD}$ | 99.30% | 100% | 98.54% | 100% |
| Hebrew$_{HTB}$ | 98.26% | 99.89% | 97.20% | 99.89% |
| Tamil$_{TTB}$ | 99.50% | 100% | 98.67% | 100% |
| Uyghur$_{UDT}$ | 97.80% | 100% | 97.19% | 100% |
| Wolof$_{WTF}$ | 97.86% | 99.95% | 97.25% | 99.95% |
| English$_{EWT}$ | 98.73% | 99.98% | 98.18% | 99.98% |
| Macro average | 96.01% | 99.04% | 93.25% | 99.04% |

Table 9: Full tree coverage for each encoding on the linguistically-diverse set of treebanks.

| Treebank | B | B-2P | 4bit | 7bit |
|---|---|---|---|---|
| Belarusian$_{HSE}$ | 96.36% | 99.95% | 96.22% | 99.95% |
| Galician$_{TreeGal}$ | 92.90% | 99.80% | 92.60% | 99.80% |
| Lithuanian$_{HSE}$ | 88.97% | 99.62% | 88.97% | 99.62% |
| Old-East-Slavic$_{RNC}$ | 72.15% | 97.75% | 72.05% | 97.75% |
| Marathi$_{UFAL}$ | 97.63% | 100% | 97.42% | 100% |
| Welsh$_{CCG}$ | 98.88% | 100% | 98.88% | 100% |
| Tamil$_{TTB}$ | 99.50% | 100% | 98.67% | 100% |
| Macro average | 92.34% | 99.59% | 92.12% | 99.59% |

Table 10: Full tree coverage for each encoding on the low-resource set of treebanks.

Tables 11 and 12 show the total number of labels needed to encode the training set for each encoding and treebank, when considering full labels (i.e., the number of combinations of syntactic labels and dependency type labels). This can be relevant for implementations that generate such combinations as atomic labels (in our implementation, label components are generated separately instead).

| Treebank | B | B-2P | 4bit | 7bit |
|---|---|---|---|---|
| PTB | 1216 | 1233 | 396 | 408 |
| Russian$_{GSD}$ | 802 | 961 | 400 | 614 |
| Finnish$_{TDT}$ | 1054 | 1223 | 435 | 685 |
| Anc-Greek$_{Perseus}$ | 1469 | 2401 | 304 | 1167 |
| Chinese$_{GSD}$ | 804 | 912 | 321 | 406 |
| Hebrew$_{HTB}$ | 754 | 798 | 317 | 357 |
| Tamil$_{TTB}$ | 262 | 274 | 153 | 164 |
| Uyghur$_{UDT}$ | 553 | 683 | 353 | 475 |
| Wolof$_{WTF}$ | 585 | 643 | 318 | 382 |
| English$_{EWT}$ | 1089 | 1281 | 487 | 709 |
| Macro average | 858.8 | 1040.9 | 348.4 | 536.7 |

Table 11: Unique labels generated when encoding the training sets of the linguistically-diverse set of treebanks, including dependency types as a component of the labels.

| Treebank | B | B-2P | 4bit | 7bit |
|---|---|---|---|---|
| Belarusian$_{HSE}$ | 1136 | 1479 | 477 | 926 |
| Galician$_{TreeGal}$ | 512 | 601 | 270 | 376 |
| Lithuanian$_{HSE}$ | 398 | 432 | 256 | 306 |
| Old-East-Slavic$_{RNC}$ | 910 | 1181 | 378 | 715 |
| Marathi$_{UFAL}$ | 275 | 291 | 197 | 223 |
| Welsh$_{CCG}$ | 474 | 514 | 265 | 312 |
| Tamil$_{TTB}$ | 262 | 274 | 153 | 164 |
| Macro average | 566.7 | 681.7 | 285.1 | 431.7 |

Table 12: Unique labels generated when encoding the training sets of the low-resource set of treebanks, including dependency types as a component of the labels.

## B Hyperparameters

We did not perform hyperparameter search, but just used MaChAmp's defaults, which can be seen in Table 13.

| Parameter | Value |
|---|---|
| dropout | 0.1 |
| max input length | 128 |
| batch size | 8 |
| training epochs | 50 |
| optimizer | adam |
| learning rate | 0.0001 |
| weight decay | 0.01 |

Table 13: Hyperparameter settings

## C Coverage Differences

It is worth noting that, while the 7-bit encoding has exactly the same coverage as the 2-planar bracketing encoding (see Tables 1, 8, 9, 10); the 4-bit encoding has less coverage than the basic bracketing. As mentioned in the main text, both have full coverage of projective trees, but there are subtle differences in how they behave when they are applied to non-projective trees. We did not enumerate all of these differences in detail for space reasons. In particular, they are the following:

- Contrary to basic bracketing, the 4-bit encoding needs to encode the arc originating from the dummy root explicitly. This means that it cannot encode non-projective, but planar trees where the dummy root arc crosses a right arc (or equivalently, the syntactic root is covered by a right arc).

- In the basic bracketing, a dependency involving words $w_i$ and $w_j$ $(i < j)$ is not encoded in the labels of $w_i$ and $w_j$, but in the labels of $w_{i+1}$ and $w_j$ (see (Strzyz et al., 2019)), as a technique to alleviate sparsity (in the particular case of that encoding, it guarantees that the worst-case number of labels is linear, rather than quadratic, with respect to sentence length). In the 2-planar, 4- and 7-bit encodings, this is unneeded so dependencies are encoded directly in the labels of the intervening words.

- Contrary to basic bracketing, in the 4-bit encoding a single / or \ element is shared by several arcs. Thus, if an arc cannot be successfully encoded due to unsupported non-projectivity, the problem can propagate to sibling dependencies. In other words, due to being more compact, the 4-bit encoding has less redundancy than basic bracketing.

## D   Plane Assignment

The 2-planar and 7-bit encodings need a strategy to partition trees into two planes. We used the second-plane-averse strategy based on restriction propagation on the crossings graph (Strzyz et al., 2020). It can be summarized as follows:

1. The crossings graph is defined as an undirected graph where each node corresponds to an arc in the dependency tree, and there is an edge between nodes $a$ and $b$ if arc $a$ crosses arc $b$ in the dependency tree.

2. Initially, both planes are marked as allowed for every arc in the dependency tree.

3. The arcs are visited in the order of their right endpoint, moving from left to right. Priority is given to shorter arcs if they have a common right endpoint. Once sorted, we iterate through the arcs.

4. Whenever we assign an arc $a$ to a given plane $p$, we immediately propagate restrictions in the following way: we forbid plane $p$ for the arcs that cross $a$ (its neighbors in the crossings graph), we forbid the other plane $(p')$ for the neighbors of its neighbors, plane $p$ for the neighbors of those, and so on.

5. Plane assignment is made by traversing arcs. For each new arc $a$, we look at the restrictions and assign it to the first plane if allowed, otherwise to the second plane if allowed, and finally to no plane if none is allowed (for non-2-planar structures).

## E   Hexatagging

Amini et al. (2023) use an intermediate representation, called binary head trees, that acts as a proxy between dependency trees and hexatags. These trees have a structure akin to binary constituent trees in order to apply the tetra-tagging encoding (Kitaev and Klein, 2020). In addition, non-terminal intermediate nodes are labeled with 'L' or 'R' based on whether the head of the constituent is on its left or right subtree. We direct the reader to the paper for specifics. However, a mapping between projective dependency trees and this structure can be achieved by starting at the sentence's root and conducting a depth-first traversal of the tree. The arc representation components for each hexatag encode: (i) the original label corresponding to the tetratag, and (ii) the value of the non-terminal symbol in the binary head tree.

## F   Speed comparison

Table 14 compares the speed of the models over an execution on a single CPU.[7] It is important to note that while SuPar is an optimized parser, in this context, we used MaChAmp as a general sequence labeling framework without specific optimization for speed. With a more optimized model, practical processing speeds in the range of 100 sentences per second on CPU or 1000 on a consumer-grade GPU should be achievable (cf. the figures for sequence-labeling parsing implementations in (Anderson and Gómez-Rodríguez, 2021)).

## G   Non-Surjectivity in Decoding

As mentioned in the main text, all encodings explored in this paper are non-surjective, meaning that there are label sequences that do not correspond to a valid tree. In these cases, the labels

---

[7]Intel(R) Core(TM) i9-10920X CPU @ 3.50GHz.

| Treebank | biaffine | 6tg | 4bit | 7bit |
|---|---|---|---|---|
| Penn-Treebank | 28.34 | 14.65 | 14.28 | 14.42 |
| UD-Russian-GSD | 28.15 | 14.27 | 14.63 | 14.30 |
| UD-Finnish-TDT | 34.68 | 18.22 | 17.56 | 17.82 |
| UD-Ancient-Greek-Perseus | 24.12 | 12.53 | 12.93 | 12.15 |
| UD-Chinese-GSD | 22.64 | 10.78 | 11.05 | 10.86 |
| UD-Hebrew-HTB | 27.06 | 13.46 | 13.15 | 13.71 |
| UD-Tamil-TTB | 29.19 | 11.98 | 12.17 | 12.87 |
| UD-Uyghur-UDT | 34.87 | 18.69 | 18.01 | 18.93 |
| UD-Wolof-WTF | 28.14 | 12.61 | 12.31 | 12.61 |
| UD-English-EWT | 35.02 | 20.03 | 19.87 | 20.17 |

Table 14: Speed (sentences per second) for the linguistically-diverse test sets.

| Treebank | 6tg | 4-bit | 7-bit |
|---|---|---|---|
| PTB | 4.01% | 8.24% | 4.13% |
| Russian$_{GSD}$ | 14.42% | 19.34% | 16.57% |
| Finnish$_{TDT}$ | 3.75% | 10.01% | 8.84% |
| Anc-Greek$_{Perseus}$ | 12.66% | 20.08% | 18.81% |
| Chinese$_{GSD}$ | 12.31% | 22.06% | 21.81% |
| Hebrew$_{HTB}$ | 10.82% | 16.76% | 16.79% |
| Tamil$_{TTB}$ | 29.06% | 36.12% | 37.67% |
| Uyghur$_{UDT}$ | 18.13% | 22.19% | 18.52% |
| Wolof$_{WTF}$ | 30.01% | 42.15% | 50.54% |
| English$_{EWT}$ | 4.01% | 12.24% | 6.48% |
| Macro average | 13.92% | 20.92% | 20.02% |

Table 15: Percentage of trees in the linguistically-diverse test sets where the label sequence output by the tagger does not correspond to a valid tree, and heuristics need to be applied to deal with unconnected nodes, cycles or out-of-bounds indexes.

are decoded using simple heuristics (e.g. skipping dependency creation if the stack is empty, ignoring material remaining in the stack after decoding, attaching unconnected nodes and breaking cycles). Table 15 shows data about the number of trees in the test set such that the labels output by the tagger do not directly correspond to a valid tree, and at least one of these heuristics has to be applied. Table 16 shows the same information in terms of the percentage of dependency arcs that are affected by said heuristics.

| Treebank | 6tg | 4-bit | 7-bit |
|---|---|---|---|
| PTB | 0.531% | 0.941% | 0.566% |
| Russian$_{GSD}$ | 0.930% | 1.479% | 1.200% |
| Finnish$_{TDT}$ | 0.291% | 0.987% | 0.780% |
| Anc-Greek$_{Perseus}$ | 0.563% | 2.291% | 1.917% |
| Chinese$_{GSD}$ | 0.705% | 1.622% | 1.593% |
| Hebrew$_{HTB}$ | 0.550% | 0.965% | 0.958% |
| Tamil$_{TTB}$ | 2.728% | 3.819% | 4.280% |
| Uyghur$_{UDT}$ | 2.052% | 2.801% | 2.191% |
| Wolof$_{WTF}$ | 1.853% | 3.043% | 3.868% |
| English$_{EWT}$ | 0.554% | 1.523% | 0.726% |
| Macro average | 1.075% | 1.947% | 1.807% |

Table 16: Percentage of dependency arcs in the linguistically-diverse test sets where heuristics need to be applied to deal with unconnected nodes, cycles or out-of-bounds indexes.