# OpenReview forum: "4 and 7-bit Labeling for Projective and Non-Projective Dependency Trees"
_EMNLP/2023/Conference — EMNLP 2023 Main_

### Official Review · Reviewer_JxDe · 2023-07-28

**Soundness:** 4
**Typos Grammar Style And Presentation Improvements:**

**Excitement:**

4: Strong: This paper deepens the understanding of some phenomenon or lowers the barriers to an existing research direction.

**Paper Topic And Main Contributions:**

In this short paper, the authors investigated sequence labeling-based dependency parsers. They proposed innovative schemes to encode projective and 2-planar trees using 4 bits (i.e., 16 labels) and 7 bits (i.e., 128 labels), respectively. Evaluation on multilingual treebank data demonstrated the effectiveness of their approach.






**Reasons To Accept:**

- The proposed encoding scheme is interesting, much simpler than prior bracketing-based sequence labeling approach, yet obtaining better performance

**Reasons To Reject:**

No major concerns were identified during the review. However, there are some minor concerns:

- The proposed method is not surjective and requires heuristics during decoding.
- The paper lacks a comparison of speed and performance with other state-of-the-art graph-based approaches, although this limitation is acknowledged in the limitations section.
- In terms of presentation, it would be beneficial to include a brief introduction to 2-planar trees. Many readers, including myself, might not be familiar with this type of trees. As a result, understanding the heuristic decoding strategy in the 4-bit case through the use of a stack is clear. However, for the 8-bit case, it is not immediately evident, and additional clarification might be helpful.










**Reproducibility:**

3: Could reproduce the results with some difficulty. The settings of parameters are underspecified or subjectively determined; the training/evaluation data are not widely available.

**Reviewer Confidence:**

3: Pretty sure, but there's a chance I missed something. Although I have a good feel for this area in general, I did not carefully check the paper's details, e.g., the math, experimental design, or novelty.

---

> ### Author Rebuttal · Authors · 2023-08-26
>
> Thank you for your review and suggestions to improve the paper. If the paper is accepted, it should be easy to address them with the extra page provided:
> - Speed and accuracy comparison with graph-based approaches: we agree that this is a useful addition and will include it.
> - Introduction to 2-planar trees: we agree that it can be difficult to understand without looking at the references in its current form. For space reasons, we were not able to provide more explanation in the 4-page version for submission. But with the extra page in the camera-ready version, we will expand the explanation and we believe it should be easy to follow, as these concepts are not very long or difficult to explain.

---

### Official Review · Reviewer_U1jy · 2023-08-03

**Soundness:** 4

**Excitement:**

4: Strong: This paper deepens the understanding of some phenomenon or lowers the barriers to an existing research direction.

**Paper Topic And Main Contributions:**

This paper offers a new approach for dependency parsing as tagging. The linearization results in 16 distinct tags for parsing projective trees and can be extended to 128 tags for parsing non-projective trees. The paper offers linear time algorithms to convert dependency trees to tags and tags to dependency trees. Comparison to 2 baselines (based on previous taggers with unbounded tag-set size) shows that their tagger achieves a very good performance while having a bounded tag set.

**Questions For The Authors:**

- I think the tree in Figure 2 can be parsed with the 4-bit schema and we do not necessarily **need** the 7-bit schema. Is that right? If yes, it would be helpful for the reader to replace that with an example that we actually need the 7-bit schema.
- The plane assignment is not really understandable given only the context in section 3. Could you elaborate more on this point?

**Reasons To Accept:**

The paper is very well written, and the proposed encoding of dependency trees to a sequence of tags is intuitive and easy to understand. Furthermore, their approach is arguably more interpretable than Hexatagging, as the tags directly correspond to the structure of the dependency tree (and not some intermediate constituency structure). The approach presented in the paper can also be generalized to parse non-projective trees.

**Reasons To Reject:**

- If I understand correctly, at inference time the authors do greedy decoding and heuristically resolve cases where the outputted tag sequence is not valid. I am wondering if it would be possible to modify the decoding algorithm to find the maximum scoring tag sequence which is also a valid one (that can cost extra compute of course).
- On the same note, the consequence of non-surjectivity on performance is not discussed in the paper. Can you provide some statistics on how many of the tags result in decoding errors and how this affects the overall performance of the parser?
- It seems necessary to compare the performance to the state-of-the-art parser as a point of reference. The two baselines provided are both taggers.
- [suggestion]: I understand that this work was probably done parallel to Hexatagging, but it would be helpful for practitioners if the authors could compare their method to Hexatagging for the camera-ready version.



**Reproducibility:**

5: Could easily reproduce the results.

**Reviewer Confidence:**

4: Quite sure. I tried to check the important points carefully. It's unlikely, though conceivable, that I missed something that should affect my ratings.

---

> ### Author Rebuttal · Authors · 2023-08-26
>
> Thank you for the review and useful suggestions.
>
> Responses to the reasons to reject (RR) and questions (Q):
>
> - RR1: As you mention, we perform greedy decoding and heuristically resolve cases where the sequence of tags does not directly map to a tree. We were especially interested in this case because it allows the user to use any off-the-shelf sequence labeling library without any special adjustments, and change such library without effort if e.g. better ones come out. We think this is interesting for practitioners and to make parsing more broadly applicable (this is mentioned briefly in the limitations section). However, the problem of designing a task-specific decoding algorithm that will take the properties of the problem into account to find the maximum-scoring tag sequence that corresponds to a valid tree is also very interesting, both from a theoretical point of view and to really squeeze the maximum possible accuracy from the framework, and we would like to tackle it in future work.
> - RR2: Yes, we believe this is a useful improvement to the paper and will provide the statistics in the extra page if the paper is accepted.
> - RR3: We will include this comparison with a state-of-the-art biaffine parser in the final version.
> - RR4: Indeed, the work was simultaneous. But we also thought that this would be interesting, so we had already started implementing hexatagging under the same framework as our encodings (for homogenous comparison) and hope to have the results ready to provide this comparison for the camera-ready version.
> - Q1: Yes, this is true – since the arcs that cross in the example are in opposite directions, the 4-bit encoding would suffice even if the tree is not projective. We will replace it with an example that really needs the 7-bit encoding.
> - Q2: In short, the plane assignment strategy that we used to partition arcs into planes for training works as follows (we will provide an explanation in the final version if the paper is accepted, and will extend the explanations of 2-planarity-related concepts in general):
>   1. The crossings graph is defined as an undirected graph where each node corresponds to an arc in the dependency tree, and there is an edge between nodes *a* and *b* if arc *a* crosses arc *b* in the dependency tree.
>   2. Initially, both planes are marked as allowed for every arc in the dependency tree.
>   3. Whenever we assign an arc a to a given plane *p*, we immediately propagate restrictions in the following way: we forbid plane *p* for the arcs that cross *a* (its neighbors in the crossings graph), we forbid the other plane (*p'*) for the neighbors of its neighbors, plane *p* for the neighbors of those, and so on.
>   4. Plane assignment is made by traversing arcs. For each new arc *a*, we look at the restrictions and assign it to the first plane if allowed, otherwise to the second plane if allowed, and finally to no plane if none is allowed (for non-2-planar structures).

---

### Official Review · Reviewer_SmyL · 2023-08-05

**Soundness:** 5

**Excitement:**

4: Strong: This paper deepens the understanding of some phenomenon or lowers the barriers to an existing research direction.

**Paper Topic And Main Contributions:**

This paper proposes two novel labeling schemes for dependency parsing. With these label sets, parsing can be casted as a sequence labeling task. The first label set uses 4 bits per word describing whether the word is a left or right dependent, whether it is an outmost dependent, and whether the word has left or right dependents. The paper further proposes a 7-bit extension set that expands the coverage to 2-Planar non-projective trees. Experiments show that the proposed label sets support good coverage on a diverse set of treebanks, and higher parsing LAS scores than baseline labeling schemes.

**Reasons To Accept:**

* Parsing as tagging is a very promising technique for fast dependency parsing. This paper introduces a highly-intuitive and yet small and bounded label set that uses only 4 bits to label each word in a sentence. This label set is slightly larger than the 3-bit label set proposed by Amini et al. (2023), but is arguable more intuitive and native to describing dependency trees. The corresponding decoding algorithm is also easy to understand and seems straightforward to implement.
* In addition to the 4-bit label set targeting at projective trees, the paper further introduces a 7-bit extension that covers 2-Planar non-projective trees which has over 99% empirical coverage on all tested treebanks (including Ancient Greek, which is highly non-projective).
* The empirical results are quite promising. The parsers using the proposed 7-bit label set outperforms the baseline label sets by a wide margin.
* The paper is quite well-written and very easy to follow. I especially appreciate the explanation on coverage (why the 4-bit label set works beyond projective trees, though designed for projective coverage).

**Reasons To Reject:**

I don't see any strong reasons.

A minor reason: the paper doesn't include comparisons with other types of dependency parsers (e.g,. the popular biaffine parser), so it is unclear whether the proposed approach performs lacks in accuracy as compared to those. It would also be helpful to include parsing speed comparisons, since fast inference should be a major selling point for parsing-as-tagging approaches. Though given the theoretical focus of this paper, I consider this point to be minor.

**Reproducibility:**

4: Could mostly reproduce the results, but there may be some variation because of sample variance or minor variations in their interpretation of the protocol or method.

**Reviewer Confidence:**

4: Quite sure. I tried to check the important points carefully. It's unlikely, though conceivable, that I missed something that should affect my ratings.

**Typos Grammar Style And Presentation Improvements:**

I recommend including speed comparisons with popular parsers, since this could be a major advantage of using parsing-as-tagging approaches.

Line 066, I recommend changing b_i to b_j, since the index i has been used to describe word w_i.

The end of section 2 talks about what happens when the label set cannot be mapped to a valid tree. It would be interesting to include some statistics on how often this happens in empirical parsing experiments.

---

> ### Author Rebuttal · Authors · 2023-08-26
>
> Thank you for your review and useful suggestions. We plan to address them with the extra page which is provided if the paper is accepted, and will improve the paper:
> - Comparison to other types of dependency parsers, such as biaffine: we will include this in the camera-ready version if accepted, including also speed comparison.
> - Typo: will be fixed.
> - Statistics on non-surjectivity (cases where the input does not map to a valid tree and heuristics are needed): we think this is interesting and we will gather and include them.

---

### Meta-Review · Area_Chair_cZk2 · 2023-09-19

**Recommendation:** 5

**Metareview:**

This paper proposes a 4-bit labeling scheme for projective dependency parsing as sequence labeling and extends it to a 7-bit labeling scheme for non-projective dependency parsing.

All the reviewers are positive, believing that the proposed scheme is simpler and more intuitive than previous schemes, the experimental results are good, and the paper is well-written. They also provide some suggestions for improving the paper, e.g., comparison of speed, comparison with other types of parsers, and clarification of section 3, which the authors have promised to address in the camera-ready.

---

### Decision · Program_Chairs · 2023-10-07

**Decision:**

Accept-Main

**Comment:**

This paper proposes a 4-bit labeling scheme for projective dependency parsing as sequence labeling and extends it to a 7-bit labeling scheme for non-projective dependency parsing.

All the reviewers are positive, believing that the proposed scheme is simpler and more intuitive than previous schemes, the experimental results are good, and the paper is well-written. They also provide some suggestions for improving the paper, e.g., comparison of speed, comparison with other types of parsers, and clarification of section 3, which the authors have promised to address in the camera-ready.